# Biocontrol potential of endophytic *Pseudomonas* strain IALR1619 against two *Pythium* species in cucumber and hydroponic lettuce

**B. Sajeewa Amaradasa**[1]*, **Chuansheng Mei**[1], **Yimeng He**[1], **Robert L. Chretien**[1], **Mitchell Doss**[2], **Tim Durham**[3], **Scott Lowman**[1]

1 The Institute for Advanced Learning and Research, The Plant Endophyte Research Center, Danville, VA, United States of America, 2 School of Plant and Environmental Sciences–Virginia Tech at The Institute for Advanced Learning and Research, Controlled Environment Agriculture Innovation Center, Danville, VA, United States of America, 3 Division of Science and Technology, Agriculture Program, School of Undergraduate Studies, Ferrum College, Ferrum, VA, United States of America

* sajeewa.amaradasa@ialr.org

**Data Availability Statement:** GenBank deposited sequences can be found at https://www.ncbi.nlm.

## Abstract

The use of fungicides to manage disease has led to multiple environmental externalities, including resistance development, pollution, and non-target mortality. Growers have limited options as legacy chemistry is withdrawn from the market. Moreover, fungicides are generally labeled for traditional soil-based production, and not for liquid culture systems. Biocontrol agents for disease management are a more sustainable and environmentally friendly alternative to conventional agroprotectants. *Pythium ultimum* is a soil borne oomycete plant pathogen with a broad taxonomic host range exceeding 300 plants. Cucumber seedlings exposed to *P. ultimum* 1 day after a protective inoculation with bacterial endophyte accession IALR1619 (*Pseudomonas* sp.) recorded 59% survival; with the control assessed at 18%. When the pathogen was added 5 days post endophyte inoculation, 74% of the seedlings treated survived, compared to 36% of the control, indicating a longer-term effect of IALR1619. Under hydroponic conditions, IALR1619 treated leaf type lettuce cv. 'Cristabel' and Romaine cv. 'Red Rosie' showed 29% and 42% higher shoot fresh weight compared to their controls, respectively. Similar results with less growth decline were observed for a repeat experiment with IALR1619. Additionally, an experiment on hydroponic lettuce in pots with perlite was carried out with a mixture of *P. ultimum* and *P. dissotocum* after IALR1619 inoculation. The endophyte treated 'Cristabel' showed fresh weight gain, but the second cultivar 'Pensacola' yielded no increase. In summary, the endophyte IALR1619 provided short term as well as medium-term protection against *Pythium* blight in cucumber seedlings and may be used as an alternative to conventional fungicides in a greenhouse setting. This study also demonstrated the potential of ALR1619 as a biocontrol agent against *Pythium* blight in hydroponic lettuce.

nih.gov/ All other relevant data are within the manuscript and its Supporting Information files.

**Funding:** This research was partially funded by USDA Specialty Crop Block Grant Program (https://www.ams.usda.gov/services/grants/scbgp) 2021B-570 for Utilizing Endophytes to Promote Hydroponic Vegetable Growth and Increase Profitability. The funders did not play any role in the study design data collection and analysis, decision to publish, or preparation of the manuscript.

**Competing interests:** The authors have declared that no competing interests exist.

## Introduction

Cucumber (*Cucumis sativus* L.) and lettuce (*Lactuca sativa* L.) are two economically important vegetables with a ubiquitous presence in both field and greenhouse environments [1–3]. *Pythium ultimum* is a soil borne oomycete plant pathogen with a broad host range exceeding 300 diverse plants [4–6]. It is capable of causing serious losses in cucumber and other cucurbitaceous crops [7,8], typically resulting in damping-off and death at the seedling stage [9–11].

Lettuce is a popular leafy crop in soilless hydroponic production systems. It is less susceptible to diseases than hydroponic spinach and has a shorter seedling to harvest window compared to fruit bearing crops like tomatoes and cucumber [12]. However, root rot caused by various *Pythium* species pose a threat in hydroponic systems since pathogen propagules can easily spread with circulating nutrient solutions and infect plants [9,10,12,13]. Several *Pythium* species have been reported to cause root rot and yield reduction in hydroponic lettuce. These include *P. aphanidermatum*, *P. dissotocum* [12,14–16], *P. ulitmum* [10,17], *P. aquatile*, and *Pythium* group F [9]. A study performed in hydroponic crops in South Africa identified various other *Pythium* species from nutrient solutions, substrates, and water sources [18]. Select *Pythium* species in lettuce from that study included *P. coloratum*, *P. irregulare*, and *Pythium* group G, HS, and F.

The use of fungicides to manage diseases has led to resistance development [19], environmental pollution [20], and death of non-target organisms [21]. The highly effective broad-spectrum soil fumigant methyl bromide (MeBr) has been restricted due to its status as a stratospheric ozone depleter [22,23]. Hydroponic crops have even more limited options since registered fungicides are typically labeled for traditional soil-based production and not for liquid culture systems [12–14,24]. Although it was initially thought that soilless hydroponic systems could provide a disease-free environment for crops, it was evident that zoosporic pathogens and other non-significant field pathogens could pose a threat to plant health and cause yield reduction even with subclinical infection [25–29]. Excessive nitrogen and a lack of natural antagonistic organisms in hydroponic systems may also be contributing factors to high disease incidence [9]. The use of biocontrol agents (BCA), including endophytes for disease control is a more sustainable and environmentally friendly alternative to conventional fungicides [30,31]. The Institute for Advance Learning and Research (IALR) maintains a sizeable collection of bacterial endophytes isolated from wild plants (n ≈ 2000). A subset has already been characterized for biocontrol and growth promotion activity in vitro using different assays.

This study provides evidence of endophyte IALR1619's positive biocontrol activity against *P. ultimum* in cucumber seedlings, and *P. ultimum* and *P. dissotocum* in hydroponic lettuce. The objectives of this study were: 1) to test the biocontrol activity of promising bacterial endophyte IALR1619 against damping-off in cucumber seedlings; 2) to ascertain disease control potential of endophytes IALR1580 and 1619 against *Pythium* blight in hydroponic lettuce cultivars; and 3) to test if hydroponic lettuce plants inoculated with a growth promoting endophyte could confer protection against the adverse effects of *Pythium* blight.

## Materials and methods

### Screening of endophytic bacteria against *Pythium* species in vitro

IALR maintains its endophyte collection in glycerol stocks. From late 2018 to mid-2019, 355 bacterial endophytes were screened in vitro against a *Pythium* isolate from soybean. Dual-culture assay was used to determine biocontrol suitability. An actively growing pathogen plug 6 mm in diameter was placed at the center of a 100-mm-diameter Petri plate with quarter strength PDA (qPDA). Ten microliters (10 μl) of an endophyte cultured in LB broth (IBI

Scientific, Dubuque, IA, USA) were inoculated at the periphery. Each plate was inoculated with four different endophytes at an equal distance to rapidly screen the aforementioned 355 bacteria for biocontrol activity. Plates were incubated at 28˚C for two days. Those bacteria which produced an at least 5 mm inhibition zone were recorded as potential endophytes having anti-*Pythium* activity. Two strains, IALR 1580 and 1619 from the aforementioned screening, were used in this study. They were tested against *P. ultimum* T89 and *P. dissotocum* 10F using the dual-culture assay described above. However, only a single endophyte was inoculated at four peripheral corners of the Petri plate. The growth inhibition of each pathogen was compared to a control plate inoculated with the pathogen, but without the endophyte. Each assay was replicated three times. When the control mycelial mat reached the edge of the plate, the average radius of the pathogen mycelial mat in endophyte inoculated plate was recorded. The radii of endophyte+ and endophyte- plates were used to calculate the inhibition %. The formula for pathogen inhibition % is expressed as:

$$\frac{[(\text{radius of mycelial mat in control plate} - \text{radius of mycelial mat in endophyte inoculated plate})]}{\text{radius of the control}} \times 100$$

## Study/Experiment overview

From July 2021 to January 2023, we conducted five experiments with endophyte IALR1619 to ascertain its biocontrol activity against *Pythium* in soilless lettuce and cucumber seedlings. Plants were maintained either in a greenhouse or an indoor farming system. Of the five experiments, two were carried out on cucumber seedlings in pots with soil and maintained in an indoor farm (Exp. 1 and 2). Two experiments were conducted on hydroponic lettuce in a greenhouse with recirculating nutrients. In these two experiments, we also included a prospective biocontrol endophyte IALR1580 (see Exp. 3) and a growth promoting endophyte IALR1379 (see Exp. 4) in addition to IALR1619. The fifth experiment was done on lettuce grown indoors in pots having perlite medium (see Exp. 5). For every experiment, *P. ultimum* was used as the disease inoculum, except for lettuce in pots with perlite (Exp. 5), in which a mixture of *P. ultimum* and *P. dissotocum* were used.

## Pathogenicity tests

A verified *P. ultimum* isolate T89 was provided by Dr. A. Boudoin of Virginia Tech, Blacksburg, VA, USA. *Pythium dissotocum* 10F was isolated from a stunted hydroponic lettuce plant at IALR. The pathogenicity of the T89 isolate was tested in 5-day old seedlings of cucumber variety Marketmore 76. Eight seedlings in 3.5-inch square pots with Sun Gro Professional Growing Mix (Sun Gro Horticulture, Agawam, MA, USA) were inoculated with T89 via attachment with two mycelial plugs per plant at the stem base using a grafting clip. Mycelial plugs were cut from actively growing cultures on qPDA Petri plates. Control cucumber seedlings were attached with water agar plugs. Within one week, all pathogen inoculated seedlings died while control plants were healthy (results not shown). To test the pathogenicity of *Pythium* isolates on lettuce, deep water culture (DWC) units were transplanted with 10-day-old seedlings of three cultivars, namely Romaine type 'Red Rosie' and 'Pensacola', and leaf type 'Cristabel'. Each hydroponic unit had 5 plants from each cultivar. Two such units were used to inoculate with 10F and T89 separately. Each pathogen was cultured for 10 d in quarter strength potato dextrose broth (qPDB) in a shaking incubator at 150 RPM and maintained at 27˚C. The mycelial broth was blended and passed through a layer of cheese cloth for inoculation. Two liters of inoculum were added to the reservoir of each unit after transplanting. Afterwards, 5 L of Virginia Tech fertilizer solution (see section below) was added to each reservoir bucket and electrical conductivity (EC) was maintained at 1 ±0.1 mS cm$^{-1}$ with pH at 5.9 ±0.1. A control

unit was maintained without the pathogens. After two weeks it was evident that the roots of *Pythium* inoculated plants were stunted, with some showing browning compared to the control (results not shown). Twenty-five days after transplanting, the fresh weight of lettuce heads was recorded. The average head weight of control plants of 'Red Rosie', 'Cristabel' and 'Pensacola' were 44, 32, and 48 g, while *Pythium* T89 inoculated cultivars recorded 31.5 g, 21 g, and 32 g of head weight, respectively. Isolate 10F resulted in 32 g, 24 g, and 20 g for the above referenced cultivars. These preliminary results showed both pathogens were responsible for yield reduction in lettuce, though blight symptoms were not observed. From inoculated cucumber seedlings and lettuce cultivars, *Pythium*-like isolates were reisolated by placing root and stem pieces on water agar plates. *Pythium*-like isolates were not recovered from control plants.

## Seeds, growth media, propagation, and hydroponic units

Cucumber variety Marketmore 76 purchased from Urban Farmer (www.ufseeds.com) was used for the experiments. Cucumber seeds were planted in 3.5-inch square pots with Sun Gro Professional Growing Mix. They were kept on an indoor bench under white LED tube lights with a 24-hour photoperiod. Lettuce seeds of 'Red Rosie', 'Pensacola', and 'Cristabel' were purchased from Johnny's Selected Seeds (Fairfield, ME, USA). They were planted in 1 square inch (6.45 cm$^2$) cells of Oasis Rootcubes® Growing Medium (Kent, OH, USA) (162 cells per sheet) and placed in the greenhouse on a propagation bench from AmHydro (Arcata, CA, USA). The seeds were irrigated with water until germinated under natural light. After germination, seedlings were irrigated for 1 min every six hours with the Virginia Tech fertilizer solution adjusted to EC 1.0 ±0.1 mS cm$^{-1}$ and pH 5.9 ±0.1. The vegetative hydroponic fertilizer solution developed at Virginia Tech (Blacksburg, VA, USA) consisted of two 100x stock solutions: Stock A (Lettuce) and Stock B (Lettuce) (see [32] for solution composition). Equal amounts of Stock A and B were used to create a final dilute solution with EC 1.0 mS cm$^{-1}$. The EC was adjusted by adding nutrient solution or water. The pH was adjusted by adding 1 N $H_2SO_4$ or 1 N KOH as needed. The fertilizer solution did not account for trace mineral elements present in city water. The EC and pH were monitored daily with an Economy pH/EC Meter (Spectrum Technologies, Inc., Aurora, IL, USA) and adjusted as needed. Two lettuce experiments were conducted in the greenhouse in deep water culture (DWC) units (VEVOR Hydroponic Grow Kits purchased on www.vevor.com). The original units could hold 36 plants/unit in 4 channels. These units were modified by dividing them into half (18 plants/unit) and doubling the space between two channels. A five-gallon bucket (19.9 L) was used as the reservoir with approximately 8 L of nutrient solution with an adjusted EC of 1.5 ±0.1 mS cm$^{-1}$ and a pH of 5.9 ±0.1. The third lettuce study was done indoors in 3.5-inch square pots added with perlite. These plants were irrigated with the same nutrient solution used for plants in DWC units.

## Endophyte origin, culturing, and inoculation

The bacterial endophytes for this study were isolated from wild plants grown in Yanceyville, VA, USA. Plant parts were surface sterilized, and bacteria isolated according to the method outlined in [33]. IARL1619 was isolated from roots of common ragweed (*Ambrosia artemisiifolia*). Its 16S sequence (GenBank accession OR663661) matched 100% to a GenBank deposited accession of *Pseudomonas gessardii*. Similarly, IALR1580 was identified as a *Ps. Protegens* (GenBank accession OR663660), isolated from leaves of white vervain (*Verbena urticifolia*). Endophyte IALR1379 belonged to *Enterobacter asburiae* (GenBank accession OQ414238) and isolated from roots of the wild plant Yellow Goatsbeard (*Tragopogon dubius*). Phylogenetic analysis of the strains was performed using the neighbor-joining method with Kimura 2-parameter (K2P) model in MEGA11 program [34–36]. 16S rDNA sequences of similar

strains from peer revied publications were downloaded from GenBank ([www.ncbi.nlm.nih.gov/genbank](www.ncbi.nlm.nih.gov/genbank)) and included in the analysis. All endophytes were preserved in glycerol stocks at -80˚ C for later use. For plant inoculation, a loop of relevant endophyte was transferred to a test tube with 4 mL of sterile LB broth and incubated overnight at 30˚ C at 200 RPM. One mL of the starter culture was added to 500 mL of sterile LB medium and incubated at the same conditions overnight until its $OD_{600}$ was approximately 1.0. The number of CFU at $OD_{600} = 1$ for IALR1379, 1580, and 1619 were $4.12 \times 10^8$, $7.10 \times 10^7$, and $7.47 \times 10^8$, respectively.

Cucumber seedlings in pots of soil were inoculated 5 d after gemination when the first one to two true leaves were visible (Exp 1 and 2). Each pot had 30 mL of bacterial solution added. Control seedlings had an equal amount of LB medium added. Lettuce seedlings were inoculated with endophytes by adding 1 mL of bacterial inoculum at the base of each plant when they were one-week old. Control plants were inoculated with 1 mL of LB medium. Irrigation was halted for 24 hours for bacteria to be internalized into the vascular system and established. One week after bacterial inoculation, lettuce seedlings with 3–4 true leaves were transplanted to DWC units (Exp. 3 and 4) or perlite pots (Exp. 5).

## Pathogen inoculation

Cucumber seedlings were inoculated either by attaching a freshly cut mycelial plug of *P. ultimum* T89 to the stem base or placing three T89 inoculated grains at the base of the seedling with stem contact. Control seedlings were attached either with water agar plugs or non-inoculated sterile grains according to the relevant pathogen inoculation method. Mycelial plugs were attached to the plants using a grafting clip. Grain inoculum was prepared by inoculating T89 in sterilized steam crimped oat and incubating for 2 weeks at room temperature. It was shaken daily to avoid clumping of the colonizing pathogen.

*Pythium* inocula for lettuce was prepared as described under "Pathogenicity Test" and added to each reservoir after lettuce seedlings were transplanted to hydroponic units. Approximated 1.5 L of *P. ultimum* T89 was added per DWC unit having 6.0 L of solution. For lettuce study in perlite pots, inocula consisted of both T89 and 10F. They were prepared by culturing in qPDB and mixing them at 1:1. Seedlings were immersed in the pathogen solution for 36 h prior to transplanting.

## Atmospheric data collection

Growth conditions were captured using Onset HOBOⓇ data loggers (Bourne, MA, USA). The average temperature, relative humidity (RH) and light intensity of each experiment is highlighted in Table 1.

**Table 1. Grow conditions and average weather parameters of different experiments.**

| Experiment | Variety/Cultivar | System | Dates | Temperature (˚C) | RH[b] % | Light (µmol $m^{-2}$ $s^{-1}$) |
|---|---|---|---|---|---|---|
| 1-Cucumber in pots with soil | Marketmore 76 | Indoor | 8[th] July to 21[st] July 2021 | 25.1 | 58.6 | 11.48 |
| 2-Cucumber in pots with soil | Marketmore 76 | Indoor | 24[th] September to 11[th] October 2021 | 25.2 | 67.9 | 10.85 |
| 3-Lettuce in DWC[a] | 'Cristabel' and 'Red Rosie' | Greenhouse | 31[st] January to 15[th] March 2022 | 21.2 | 55.1 | 254.78 |
| 4-Lettuce in DWC | 'Red Rosie' | Greenhouse | 21[st] July to 29[th] August 2022 | 25.8 | 73.7 | 179.83 |
| 5-Lettuce in pots with perlite | 'Cristabel' and 'Pensacola' | Indoor | 7[th] December 2022 to 5[th] January 2023 | 20.8 | 41.0 | 32.65 |

[a]DWC, Deep water culture units.

[b]RH, relative humidity.

[https://doi.org/10.1371/journal.pone.0298514.t001](https://doi.org/10.1371/journal.pone.0298514.t001)

## Indoor and greenhouse experiments

**Experiment 1 and 2: Cucumber–*P. ultimum* pathosystem with endophyte IALR1619.**
Two experiments were carried out to ascertain biocontrol potential of IALR1619 in cucumber against *P. ultimum* T89 (see Table 1 for experimental conditions). Seedlings of variety Marketmore 76 were used in both studies. The experiments were structured as a completely randomized design (CRD). Cucumber seeds were planted in 3.5-inch square pots and kept indoors under LED lights with 24-hour photoperiod. At one to two true leaf stage, seedlings were inoculated with endophyte IALR1619 by adding 30 mL to each pot. For Experiment 1, pathogen T89 inoculation was carried out using a mycelial plug 24 h after adding the endophyte. The pathogen inoculation of Experiment 2 was carried out using grain inocula, 5 d after adding bacterial endophyte. Endophyte and pathogen inoculations are described in the above sections. Treatments for Experiment 1 consisted of plants inoculated with both endophyte and *Pythium* T89; positive control with only T89; plants with only endophyte; and negative control having no endophyte or pathogen. The first two treatments had 22 seedlings each while the 3rd and 4th treatments consisted of 21 seedlings each. Since plants inoculated only with endophytes did not show stress or growth decline compared to the negative control (Tables 1 and 2 in S2 Text), Experiment 2 was conducted only with three treatments: plants inoculated with both endophyte IALR1619 and Pythium T89; positive control having only T89; and negative control with no endophyte or pathogen. There were 27 plants for the first treatment while positive and negative controls had 28 and 30 plants, respectively. The first experiment was designed to understand if exudates or volatile compounds of IALR1619 could inhibit *P. ultimum* (short term protection) while the second experiment tested if the endophyte could protect seedlings for a comparatively longer period of time by colonizing the roots of cucumber seedlings.

**Experiment 3 and 4: Hydroponic lettuce–*P. ultimum* pathosystem with endophytes.**
For Experiment 3, two endophytes, IALR1580 and IALR1619 with biocontrol potential were used in hydroponic lettuce cultivars 'Cristabel' and 'Red Rosie'. A randomized completed block design (RCBD) with three blocks and four treatments per block was used. Two treatments had plants inoculated with one endophyte (IALR1619 or 1580) and *Pythium* T89. The third was positive control with only *Pythium*, and the fourth treatment was negative control without endophytes or pathogen. Each treatment in each block had 9 plants per cultivar. Therefore, the total experiment evaluated 27 plants from each cultivar per treatment. Since the above experiment gave good results for IALR1619, it was included in Experiment 4. Growth promoting endophyte IARL1379 was added as a new treatment. Only 'Red Rosie' was used in this experiment. Plants were spaced every other in-channel aperture to allow more space between plants. Similar to the previous experiment, there were four treatments: endophyte (IALR1619 or 1379) inoculated plants exposed to *Pythium*; positive control with *Pythium* only; and negative control without endophyte or pathogen. The full experiment had 27 'Red Rosie' plants per treatment. A growth promoting bacterium was included to test if it could assist lettuce plants evade growth decline attributed to *Pythium* T89. Growth promoting traits included auxin synthesis, N fixation, and phosphate solubilization. Auxin synthesis was quantified by modified Patten and Glick's protocol [37] published earlier by authors [38] and concentration was expressed as µg IAA mL$^{-1}$ of bacterial cells. N fixation screening followed the protocol of Pathak and Kalekar [39], which showed positive (+) or negative (–) growth on N free medium. Soluble P was quantified with Murphy and Riley [40,41] method. The IAA production of IALR1379 was 78.9 µg mL$^{-1}$ of bacterial cells whereas IALR1580 and 1619 synthesized < 5 µg of IAA per mL of cells. N fixing was positive for all three endophytes while P solubilization was 300, 368, and 161 µg mL$^{-1}$ in 6-day-old bacterial cultures for IALR1379, 1580, and 1619, respectively.

**Experiment 5: Lettuce in perlite pots–*P. ultimum* and *P. dissotocum* pathosystem with endophyte 1619.** Two lettuce cultivars, 'Cristabel', and 'Pensacola' were used in this experiment (see Table 1 for experimental conditions). Half of the seedlings were inoculated with IALR1619 while on the propagation table as described under "Endophyte Origin, Culturing, and Inoculation". Control seedlings were added with LB. Each cultivar had 13 plants per treatment. After one-week endophyte treated and untreated seedlings were submerged in *Pythium* inoculum separately for 36 h and transplanted into 3.5-inch square pots with perlite. The inoculum consisted of T89 and 10F (see "Pathogen inoculation" for details). Pots were kept on an indoor bench under white LED tube lights with 24-hour photoperiod. Plants of the two cultivars were placed separately according to a completely randomized design. Soon after transplanting, pots were added with 200 mL of Virginia Tech nutrient solution designated for hydroponic lettuce. Thereafter, 40 mL of solution was added per plant each day. The EC and pH of the nutrient solution were adjusted to 1.5 mS cm$^{-1}$ and $5.9 \pm 0.1$ before irrigation. The second endophyte inoculation was done after one-week of transplanting by adding 30 mL of IALR1619 inoculum with $OD_{600} = 1$, per pot. After 5 days plants were exposed to a second dose of *Pythium* inocula by adding 35 mL of T89 and 10F isolate mixture, which was prepared as described above.

## Lettuce growth indices

After harvesting, fresh and dry weights of lettuce heads were recorded. Root dry weights were measured. Root fresh weights were not analyzed since it was difficult to uniformly remove water from them. Additionally, lettuce head height was measured for Romaine cv. 'Red Rosie' given its innate upright architecture. For lettuce in perlite pots, only fresh and dry weights of shoots were recorded.

## Mode of action of IARL1619

**Antifungal activity of exudates and volatile organic compounds (VOCs) of strain IALR1619.** To elucidate the mode of action of IALR1619 against *Pythium* species, a plate assay was conducted after separating bacterial exudates from cells. Overnight grown bacteria in LB medium were filtered through a 0.2 µm pore size PES membrane using a bottle top filtration unit. The bacterial cells left on the filter paper were washed with sterile phosphate buffered saline (PBS) and collected to a conical tube. Dual-culture assays were conducted in triplicate with *Pythium* isolate T89 and the filtrate containing bacterial exudate; cells from membrane without exudates; and the original culture in LB; separately. The control plates were inoculated with sterile water at the periphery instead of bacteria. To test if organic volatile compounds (VOCs) have any effect on *Pythium* species, a two-sealed-bottom-plate assay was done. IALR1619 was cultured overnight in LB medium and 150 µl added to a 100 mm-diameter petri plates with LB agar. The culture was spread evenly across the plate and let sit for 30 minutes for liquid to absorb. A plug of *Pythium* T89 was placed at the periphery of a qPDA plate and carefully inverted and kept on the IALR1619 inoculated LB plate. The two bottom plates were sealed with parafilm strips to prevent bacterial VOC from escaping the plate. The experiment was done in triplicate with control plates having sterile water instead of bacteria. The same experiment was repeated for *Pythium* 10F. The plates were incubated at room temperature and observed for pathogen inhibition after two days.

**Analysis of VOCs.** IALR1619 was cultured in a 500 mL flask with 200mL of LB for 3 days at 150 RPM and 30 ˚C. The flask was covered with aluminum foil and plugged with a silicon rubber cap to prevent escape of volatile compounds. The control flask had LB medium only. The volatile compounds were extracted to a 10 mL syringe and injected into an Agilent 7890A

gas chromatograph with an Agilent 5975C Triple-Axis mass spectrometer (GC/MS) (Agilent Technologies, Santa Clara, CA, USA) for the analysis. The instrument was programmed as follows: The initial oven temperature was maintained at 35°C for 2 min and gradually increased to 190°C at a ramp rate of 4°C min⁻¹ and held for 15 min. The GC transfer line was maintained at 100°C and the detector at 250°C. The inlet pressure was at 55 kPa, He flow at 40 mL min⁻¹; and filament voltage at 70 eV of ionization energy. The VOCs were identified by comparing instrument's built-in reference spectra to sample spectra masses. The analysis was repeated once.

## Data analysis

Data analysis was performed using SAS® OnDemand for Academics (https://www.sas.com/en_us/software/on-demand-for-academics.html) software. The differences in growth inhibition of *Pythium* isolates by IALR1580 and 1619 in vitro were analyzed using the non-parametric rank-based Kruskal-Wallis test. The first two experiments with cucumber seedlings were analyzed using a Chi-square Test of Independence to infer if survivability of seedlings was associated with presence of endophyte IALR1619. The data of the experiments 3–5 were tested for normality using Shapiro-Wilk test. The variables that failed the test were again checked for normality after removing the highest and lowest extreme values. Non-normal variables were analyzed using Kruskal-Wallis test. Accordingly, Kruskal-Wallis test was used for the experiment 3 dry shoot and root weights since they were not normal. An analysis of variance (ANOVA) test was performed on other lettuce growth indices to infer endophyte treatment effect on *Pythium* species. Pairwise comparisons of treatment groups were performed by Fisher's least significant difference (LSD) procedure. The Student's t test was used to analyze treatments of lettuce plants in perlite pots.

## Results

### Phylogenetic analysis of strains and inhibition of *Pythium* species by IALR1619 and 1580 in vitro

Phylogenetic analysis of endophytes used in this study and their relationship with other similar microorganisms are depicted in Fig 1. The strains IALR1379, 1580, and 1619 clustered with respective GenBank deposited strains with a high bootstrap value of >98%. The average *Pythium* inhibition shown by the two biocontrol endophytes IALR1580 and 1619 are as follows with standard deviation (SD) in parenthesis. The endophyte IALR1619 caused 59% (± 1.39 SD) and 64% (± 0.8 SD) inhibition for *P. ultimum* and *P. dissotocum*, respectively. A lesser inhibition of 44% (± 0.66 SD) and 47% (± 0.76 SD) was observed for the same two pathogens by the bacterial strain IALR1580 (Fig 2). Statistical analysis did not result in any difference among growth inhibitions caused by the two biocontrol endophytes (S1 Fig and S1 Text).

### Indoor experiments with cucumber seedlings

The number of survived cucumber seedlings were recorded for each treatment after 10 days of *P. ultimum* inoculation. For experiment 1, 100% survivability was observed for the negative control and seedlings inoculated only with endophyte IALR1619. However, the positive control had only 4 live seedlings (18.2% survivability) while 13 IALR1619 treated seedlings survived (59.1% survivability) with no damping-off symptoms. Above two treatments and relevant seedling survivability were tested for association using Chi-square test and found significant with $\chi^2$ (1, N = 44) = 7.77, and $P$ = 0.005 (Table 3 in S2 Text). Therefore, plants treated with endophyte IALR1619 were more likely to survive than non-inoculated seedlings when

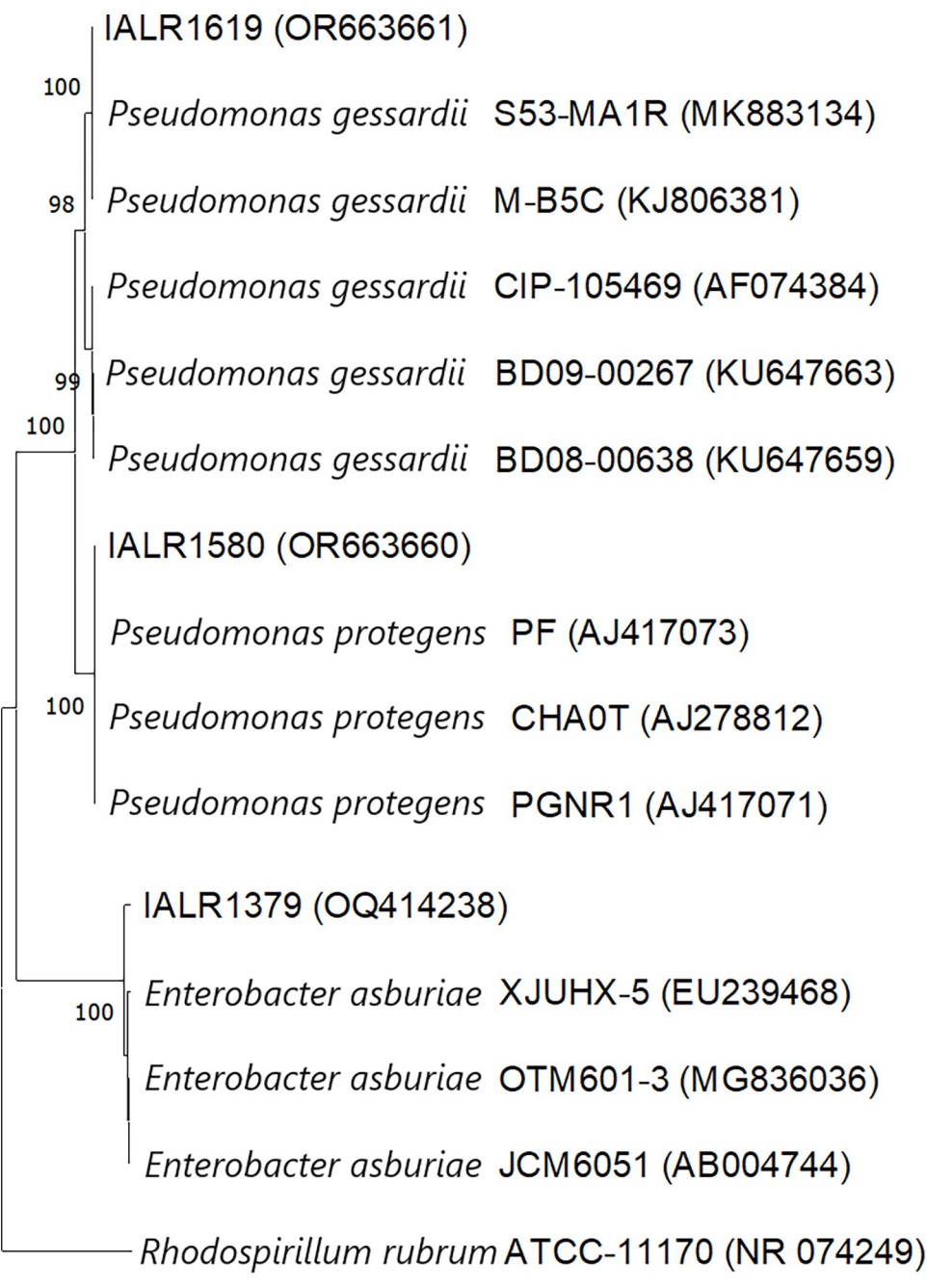

**Fig 1. Neighbor joining phylogenetic tree of bacterial endophytes IALR1379, 1580, 1619, and reference strains based on 16S rDNA sequences.** The tree is mid-point rooted. Bootstrap values generated by 500 replications are shown at the base of each branch. The GenBank accession number of each strain is in parentheses. *Rhodospirillum rubrum* is used as an outgroup. The scale bar represents approximately 2 base substitutions per 100 nucleotide positions.

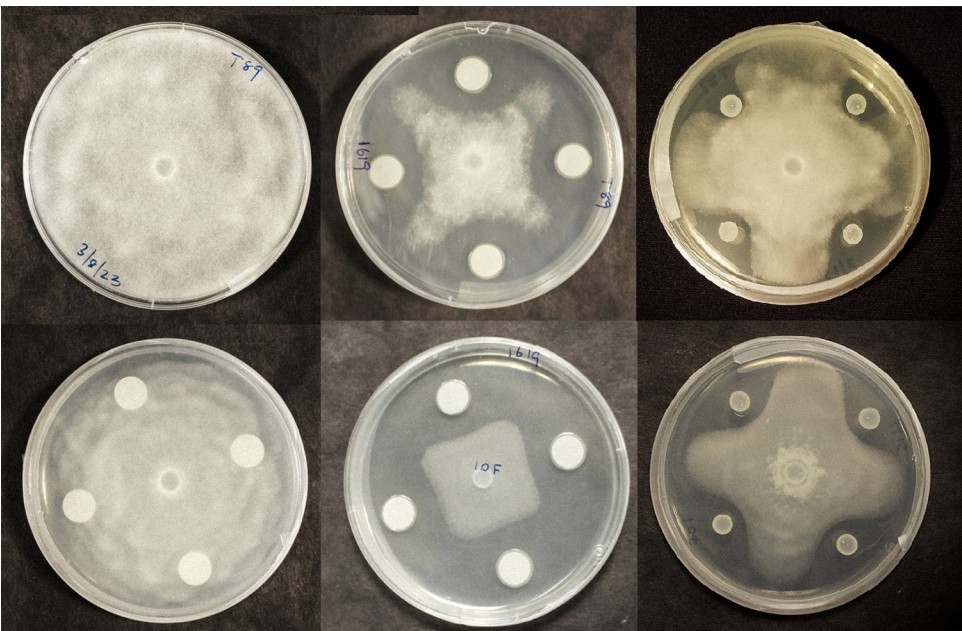

**Fig 2. Inhibition of *P. ultimum* T89 and *P. dissotocum* 10F by bacterial endophytes IALR1619 and 1580.** The top left is the control plate for T89. The top center and right plates show inhibition of T89 by IALR1619 and 1580. The bottom left plate is the control for 10F while the bottom center and right plates show inhibition of 10F by IALR1619 and 1580.

exposed to *P. ultimum*. In this experiment, pathogen inoculation was performed one day after adding the bacterial endophyte to seedlings.

Experiment 2 was carried out to determine if the endophyte 1619 could confer resistance to *Pythium* blight for a longer period. Twenty endophyte treated seedlings survived (74.1%) at the end of the experiment while only 10 were alive for the positive control (35.7% survivability). There was significant association between treatment and survivability with $\chi^2$ (1, N = 55) = 8.15, and *P* = 0.004 (Table 4 in S2 Text), hence endophyte 1619 was more likely to protect cucumber seedlings from *Pythium* infection even after 5 days of inoculation. Fig 3 shows a section of cucumber seedlings 2 days after inoculating with the pathogen.

## Greenhouse experiments with hydroponic lettuce

For the first hydroponic lettuce experiment two cultivars ('Cristabel' and 'Red Rosie') were tested for *Pythium* blight after inoculation with two potential biocontrol endophytes, separately (raw data in Table 1 in S3 Text). Shoot fresh weight and dry weight of IALR1619 treated plants were significantly higher in both lettuce cultivars compared to the positive control (Table 2 and S3 Text). 'Cristabel' treated with IALR1619 had a shoot fresh weight gain of 29% over the positive control whereas 'Red Rosie' recorded a 45% increase. Plants treated with the IALR1580 did not show any shoot weight difference compared to the positive control. The negative control having no endophytes or pathogen recorded shoot weights similar to IALR1619 treated plants in 'Cristabel'. The shoot weight of 'Red Rosie' was highest for the negative control followed by IALR1619 (Table 2).

For 'Cristabel', root dry weight was similar for IALR1619, 1580, and positive control. The negative control recorded a lower root weight than other treatments except for IALR1580 treated plants. Root weight for 'Red Rosie' was highest for the negative control followed by IALR1580 treated plants. Shoot height was recorded only for 'Red Rosie' and post hoc analysis

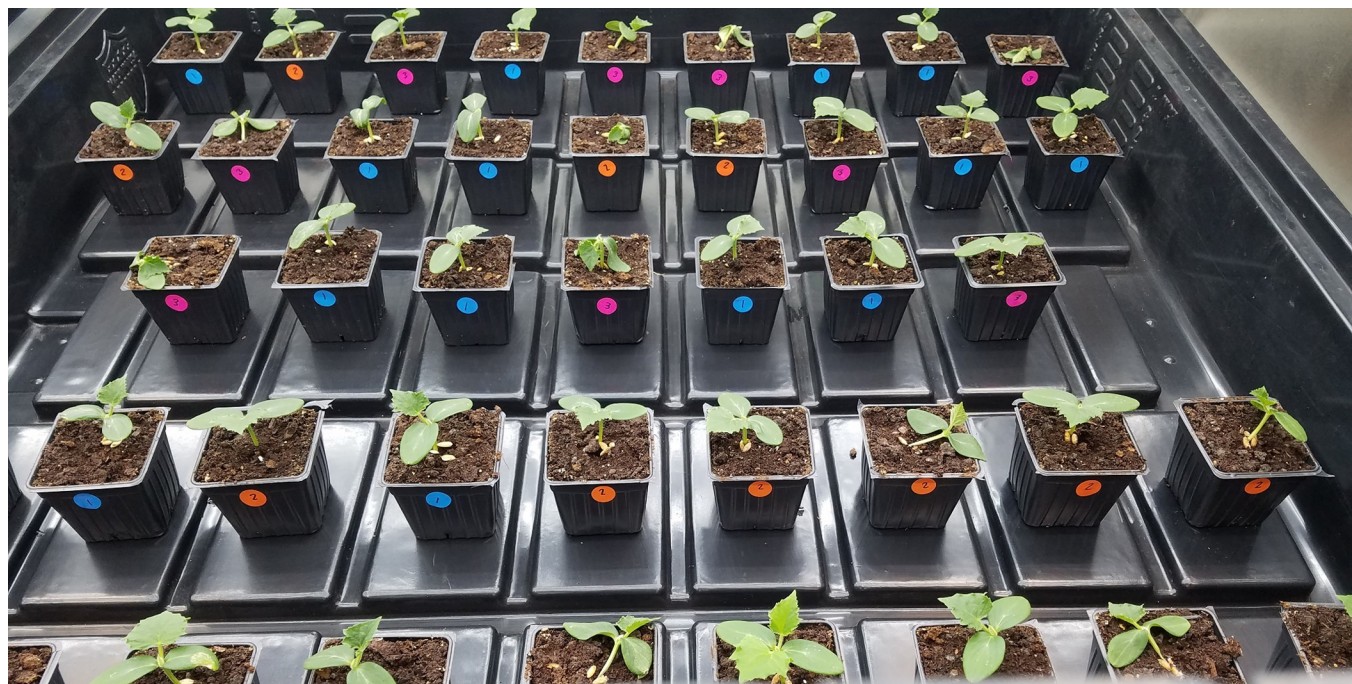

**Fig 3. Cucumber seedlings 2 days after inoculation with *P. ultimum*.** This was the second study carried out to ascertain the effectiveness of biocontrol endophyte IALR 1619 against *Pythium* blight. Pots with orange dots had been inoculated with the endophyte 1619 prior to *P. ultimum* exposure. Positive control plants without the endophyte 1619 are marked with pink dots. Negative control plants are marked with blue dots and do not show any damping off symptoms.

indicated in taller plants for negative control and no difference among the rest of the treatments (Table 2). Fig 4 shows representative plants of each 'Red Rosie' treatment at harvest.

Romaine cultivar 'Red Rosie' was used for the second DWC study with endophytes IALR1619 and 1379 against *P. ultimum* (data in Table 1 in S4 Text). The highest fresh and dry shoot weights were recorded for the negative control followed by IALR1619 treatment (Table 3). The shoot weights of the positive control were less than the IALR1619 treated plants. IALR1379 did not perform better than the positive control for shoot weight. The shoot fresh weight gain of IALR1619 treated plants was 18% higher than the positive control. Root dry weights were not significantly different for the negative control and IALR1619 treated plants. However, the negative control had higher dry root weight than both the positive control and

**Table 2. Effectiveness of endophytes IALR1619 and 1580 against *P. ultimum* in hydroponic lettuce cultivars 'Cristabel' and 'Red Rosie'.**

| Cultivar | Treatment | Shoot fresh weight (g) | Shoot dry weight (g) | Root dry weight (g) | Shoot height (cm) |
|---|---|---|---|---|---|
| Leaf Lettuce cv. 'Cristabel' | Positive control | 22.31 ± 4.72 b | 1.13 ± 0.24 b | 0.49 ± 0.08 a | Not measured |
| | Negative control | 27.72 ± 5.20 a | 1.35 ± 0.25 a | 0.44 ± 0.05 b | Not measured |
| | IALR1619 | 28.79 ± 4.90 a | 1.44 ± 0.26 a | 0.52 ± 0.07 a | Not measured |
| | IALR1580 | 19.22 ± 4.12 b | 1.04 ± 0.21 b | 0.47 ± 0.05 ab | Not measured |
| Romaine cv. 'Red Rosie' | Positive control | 17.03 ± 5.26 c | 1.07 ± 0.32 c | 0.48 ± 0.05 c | 22.43 ± 2.72 a |
| | Negative control | 33.12 ± 6.90 a | 1.93 ± 0.43 a | 0.57 ± 0.08 a | 25.33 ± 0.89 b |
| | IALR1619 | 24.64 ± 5.36 b | 1.52 ± 0.31 b | 0.48 ± 0.05 bc | 23.29 ± 1.66 a |
| | IALR1580 | 18.21 ± 5.56 c | 1.15 ± 0.35 c | 0.51 ± 0.05 b | 23.40 ± 1.90 a |

Note: Data in the table are means ± standard deviation. Means following same letter in a column for each cultivar are not significantly different at $P = 0.05$.

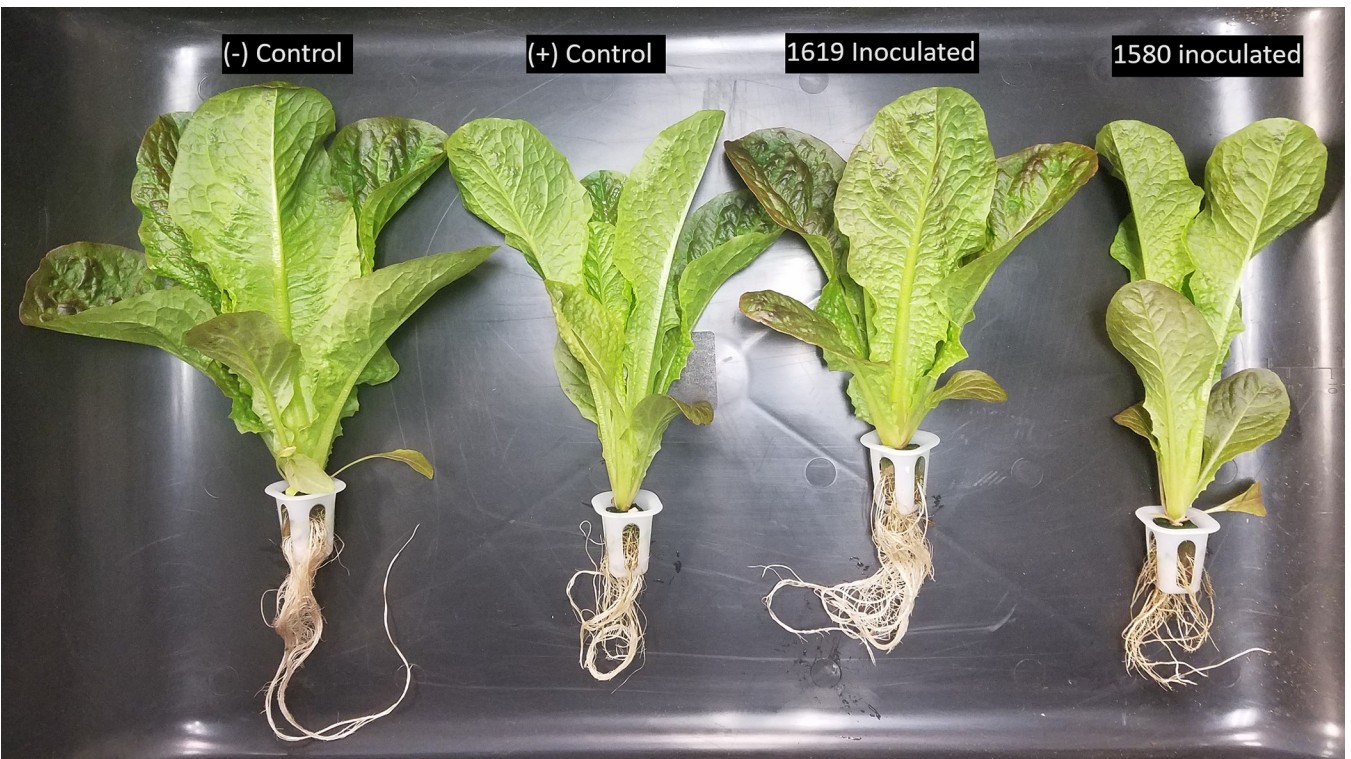

**Fig 4. Control of *Pythium* blight in hydroponic Romaine lettuce cultivar 'Red Rosie' using bacterial endophytes.** Lettuce seedlings were inoculated with endophytes IALR1619 and 1580 and maintained for one week before transplanting. At transplant *P. ultimum* T89 was added to nutrient reservoirs. Plants represent negative control, positive control, IALR1619 inoculated, and IALR1580 inoculated, at harvest.

IALR1379. The positive control and plants treated with IALR1379 and IALR1619 also did not differ from one another for root dry weight (Table 3). The negative control recorded the highest shoot height while IALR1619 and positive control did not show a difference in height. IALR1379 recorded the lowest shoot height.

## Indoor lettuce experiment in perlite pots

Lettuce cultivar 'Cristabel' treated with endophyte 1619 had a higher gain over the positive control for both dry and fresh shoot weight (Fig 5 and S5 Text). The increase in fresh harvest of 'Cristabel' inoculated with IALR1619 was 49.9% compared to the positive control. The mean fresh shoot weight of 'Pensacola' plants treated with IALR1619 was not statistically different to non-treated positive control at $P = 0.05$. However, the mean dry shoot weight of the positive control was higher than the endophyte treated plants.

**Table 3. Effectiveness of endophytes IALR1619 and 1379 against *P. ultimum* in hydroponic lettuce cultivar 'Red Rosie'.**

| Cultivar | Treatment | Shoot fresh weight (g) | Shoot dry weight (g) | Root dry weight (g) | Shoot height (mm) |
|---|---|---|---|---|---|
| Romaine cv. 'Red Rosie' | Positive control | 36.14 ± 15.05 a | 2.10 ± 0.81 c | 0.77 ± 0.12 b | 291.78 ± 33.16 a |
| | Negative control | 53.98 ± 12.56 d | 3.16 ± 0.77 a | 0.86 ± 0.13 a | 320.41 ± 24.49 c |
| | IALR1619 | 42.57 ± 12.61 c | 2.49 ± 0.62 b | 0.81 ± 0.12 ab | 290.52 ± 30.36 a |
| | IALR1379 | 23.24 ± 7.53 b | 1.78 ± 0.56 c | 0.80 ± 0.10 b | 240.22 ± 27.04 b |

Note: Data in the table are presented as means with ± standard deviation. Means following same letter/s in a column are not significantly different at $P = 0.05$.

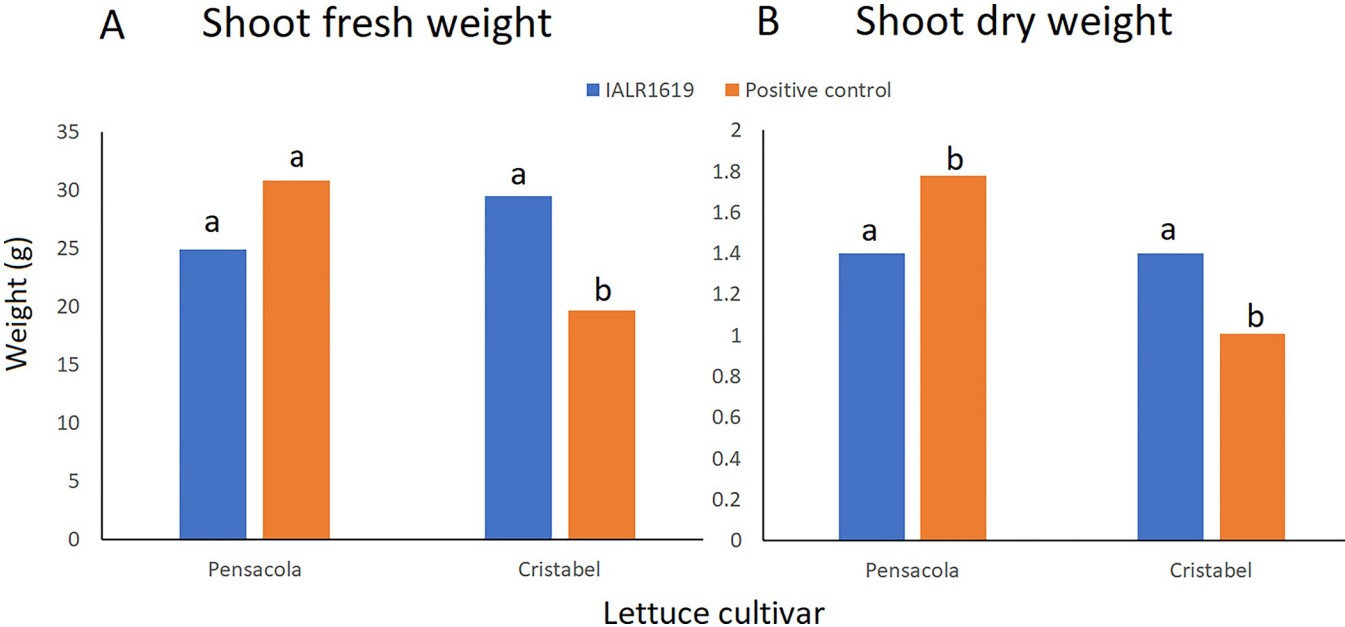

**Fig 5. The effect of endophyte IALR1619 against *Pythium* blight in hydroponic lettuce in pots with perlite.** A mixture of *P. ultimum* and *P. dissotocum* was used as the inoculum. The graph depicts the fresh shoot weight (A) and dry shoot weight (B) of lettuce cultivars 'Pensacola' and 'Cristabel'. Bars having same letter above them are not significantly different for each cultivar at *P* = 0.05.

### Effect of IALR1619 exudates and VOCs against growth of *Pythium* isolates

IALR1619 exudates without the bacterial cells did not inhibit *Pythium* growth. Both IALR1619 in LB medium and the residue bacterial cells collected after filtration of exudates did show pathogen inhibition. Two-sealed-bottom-plate assays with IARL1619 declined growth of both 10F and T89 isolates of *Pythium*. The growth inhibition started to show only on the second day and afterwards (Fig 6). After several days it was observed that mycelial mats exposed to VOCs of the endophyte 1619 were thinner than the control.

### GC/MS identification of VOCs produced by IALR1619

The analysis resulted in a single tall peak at 24.2 retention time (RT) for IALR1619 strain in both repeats (Fig 7). There was no corresponding peak in the control. The volatile compound depicted in the peak was identified as 1-undecene.

### Discussion

We screened >300 bacterial endophytes using dual-culture assays against a *Pythium* isolate *in vitro* and selected two isolates with high biocontrol potential for greenhouse studies in either cucumber and/or hydroponic lettuce. Additionally, one growth stimulating endophyte was tested in hydroponic lettuce to ascertain if it can help plants tolerate the adverse growth decline effect of *Pythium* species. *Pseudomonas* strain IALR1619 inoculated plants consistently performed better than the positive control inoculated only with *Pythium* pathogen. Though both lettuce and cucumber plants were protected by IALR1619 against *Pythium* blight, additional studies are needed to understand the level of cultivar specificity of IALR1619 within these species. Both 'Red Rosie' and 'Cristabel' cultivars of lettuce inoculated with IALR1619 produced better yield in the presence of *Pythium* species, but 'Pensacola' cultivar failed to produce

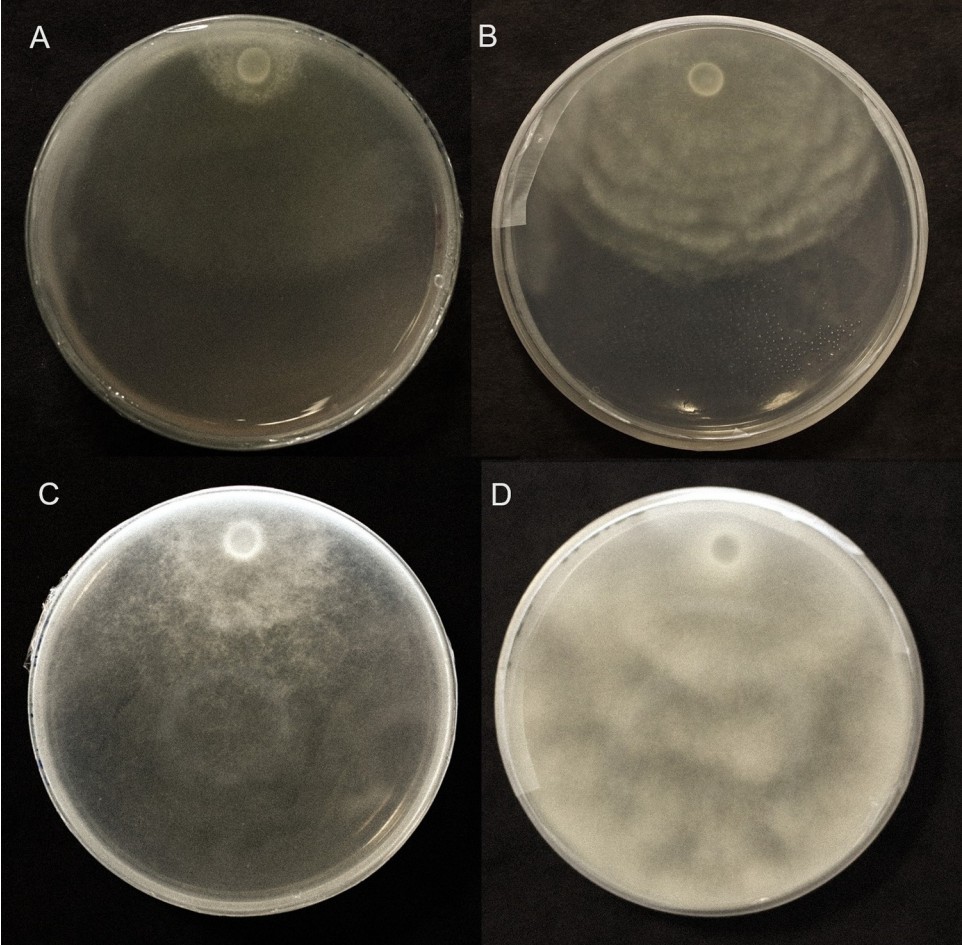

**Fig 6. The effect of volatile organic compounds (VOCs) of bacterial endophyte IALR1619 against *Pythium* species.** Two-sealed-bottom-plate assay results are shown for *P. ultimum* and *P. dissotocum*. IALR619 and *Pythium* isolates were cultured separately, afterwards, bottom plates inverted, sealed, and incubated at room temperature for 3 days. (A) The growth decline of *P. dissotocum* 10F due to VOCs and (B) the 10F control plate. (C) The thinned and inhibited mycelia of *P. ultimum* T89 due to VOCs of IALR1619 and (D) the T89 control.

significant yield over the control. Only one cucumber variety Marketmore 76 was tested against *Pythium* blight.

A previous study conducted at this institute tested the response of six hydroponically grown lettuce cultivars to a growth promoting bacterium *Ps. psychrotolerans* IALR632 [32]. The Green Oakleaf cultivar gave the best growth in response to IALR632 inoculation over the control while romaine cultivar 'Red Rosie' shoot and root growth were not significant. The above study agrees with our results showing differential performance of bacterial endophytes among lettuce cultivars although IALR1619 in this study was a biocontrol bacterium. A literature review on cultivar specificity of individual bacterial endophytes on other vegetable crops was unsuccessful. However, an analysis of bacterial endophytes using 16S rRNA-based techniques in three potato varieties revealed plant tissue (stems, roots, and tubers) and varietal endophyte specificity [42]. Another study to investigate if bacterial endophytes in seed, stem, and root tissues of cotton seedlings were influenced by host genotypes found significant cultivar differences for populations of endophytes recovered from radicle and seedling tissues [43].

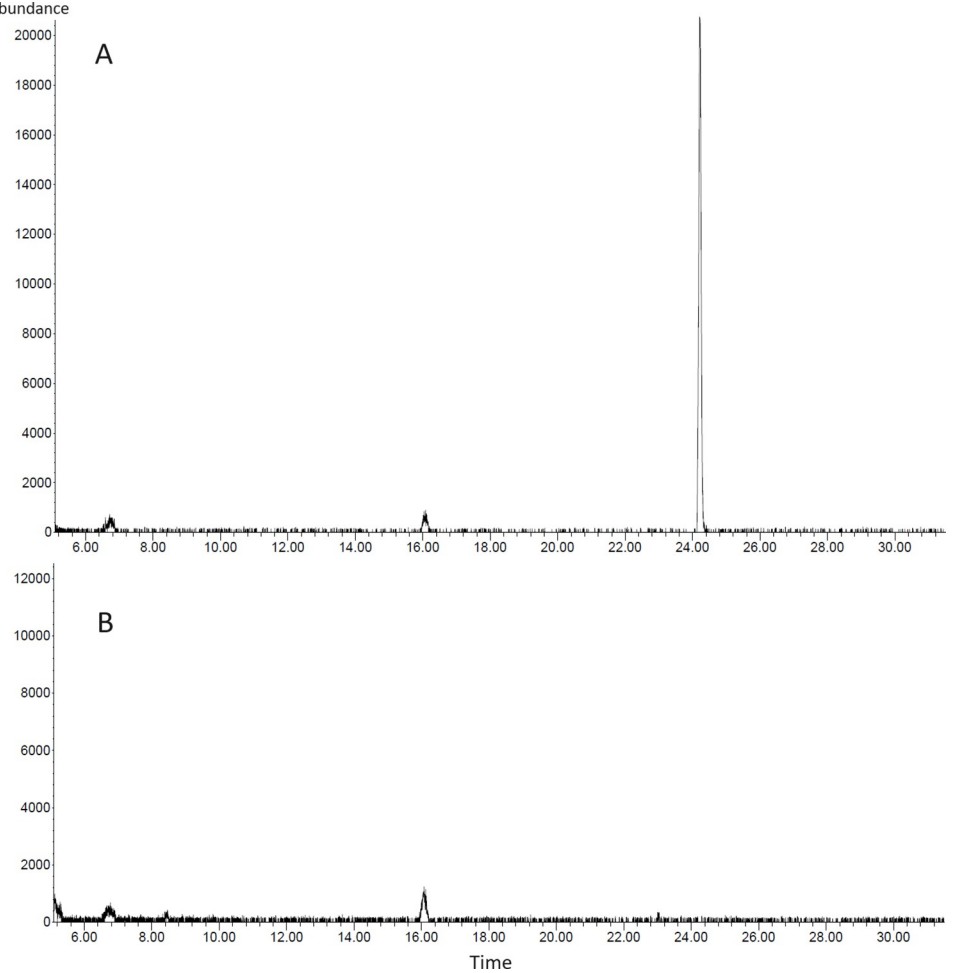

**Fig 7.** Chromatographic profiles of VOCs of (A) strain IALR1619 incubated for 72 h in LB medium and (B) uninoculated LB medium. The peak corresponds to 1-undecene.

The composition of bacterial functional groups also differed among seedlings of cotton cultivars.

The endophytes IALR1619 and 1379 were originally isolated from plant roots while IALR1580 was recovered from leaves. We inoculated test plants with endophytes by applying them to plant roots, as this is the most effective mode of colonization. Colonization of plants by bacterial endophytes has been previously investigated by researchers at this institute. In one study, GFP derivatives of *Burkholderia phytofirmans* strain PsJN were visible in leaves of switchgrass 14 days post root inoculation [44]. Another investigation reisolated *Ps. psychrotolerans* strain IALR632 from lettuce leaf samples 10 d after root inoculation [32]. The above experiments show bacterial endophytes first colonize in roots and eventually translocate into other parts of the plants. The *Pseudomonas* strain IALR11619 in this study was given 5–7 days to colonize seedlings since in our experience most endophytes have been found to behave in a similar manner.

Various bacterial endophytes have been reported to control *Pythium* blight in cucumber [45–49]. They have shown different modes of action on *Pythium* species. The bacterium *Serratia plymuthica* prevented *Pythium* blight in cucumber seedlings by stimulating defense reactions [45] while *Bacillus amyloliquefaciens* showed a broad spectrum of antifungal activity

against several plant pathogens including *Pythium aphanidermatum* in tomato by secreting different metabolites [46]. Four bacterial endophytes isolated from halophytic plant *Prosopis cineraria* showed potential to control *P. aphanidermatum* in vitro [47] and one of them inhibited the pathogen by releasing volatile organic compounds. There was a dearth of peer-reviewed articles of bacterial endophytes controlling *Pythium* decline in hydroponic lettuce. It is likely that VOCs produced by IALR1619 played an important role in controlling *Pythium* species since an inverted two-bottom-plate assay inhibited the pathogen growth. When separated from bacterial cells, bacterial exudates did not have any anti-oomycotal effect. Cucumber seedlings inoculated with *P. ultimum* just after drenching with IALR1619 showed significantly lower damping-off and mortality compared to the control. This demonstrated that VOCs produced by the endophyte had an impact on pathogen control as there was no time for the bacterium to colonize cucumber seedlings.

The GC/MS analysis of volatiles released by IALR1619 detected a single compound, 1-undecene in high abundance. This compound has been previously reported as a VOC associated with biocontrol strains of *Pseudomonas* species. A rhizopheric strain of *Pseudomonas chlororaphis* with antifungal activity against *Rhizoctonia solani*, several *Fusarium*, and *Colletotrichum* spp. was detected to produce 1-undecene as the most abundant volatile compound [50]. Another *Pseudomonas* strain with strong activity against 11 plant pathogenic fungi had 1-undecene in high quantities in its volatile compounds [51]. The volatile compounds of the said strain could inhibit the pathogen mycelial growth more than the diffusible substances in media. There may be other VOCs produced by IALR1619 but not detected due to their relatively weak signals in the presence of 1-undecene. Therefore, further investigations are needed to identify additional fungicidal or fungistatic VOCs.

Another prospective experiment is to study how effective IALR1619 is against *P. aphanidermatum*. A study done on hydroponic lettuce and spinach revealed *P. aphanidermatum* dominated when temperature of the nutrient solution was above 23°C and *P. dissotocum* commonly infected plants at lower temperatures [12]. If IALR1619 is equally effective against both *P. dissotocum* and *P. aphanidermatum*, it would be more beneficial to growers engaged in hydroponic lettuce. Though we did not test our IALR1619 for growth promotion activity *in planta*, in vitro assays indicated N fixing and P solubilizing ability for this strain.

We investigated the potential of three bacterial endophytes in controlling *Pythium* blight either in cucumber seedlings or hydroponic lettuce. It was revealed IALR 1619 is a potential biocontrol bacterium in both cucumber and hydroponic lettuce. Our results also suggest that endophytic bacteria are cultivar specific; therefore, it is important to test prospective endophytes in different cultivars within a crop to ascertain their full potential in disease control and evasion. This research sheds light on a beneficial endophyte with protective biocontrol activity against *Pythium* species in greenhouse cucumber and hydroponic lettuce.

## Supporting information

**S1 Fig. Inhibition of *P. ultimum* T89 and *P. dissotocum* 10F by bacterial endophytes IALR1619 and 1580.** Growth inhibition bars having same letter above them are not significantly different at *P* = 0.05.
(TIF)

**S1 Text. In vitro growth inhibition of *Pythium* isolates caused by endophytes IALR1580 and IALR1619.** (Tables 1 and 2) Inhibition percentages of *Pythium* 10F and T89 isolates in the presence of endophytes IALR1580 and 1619. (Table 3) Statistical analysis for inhibition of *Pythium* isolates by IALR1580 and 1619 endophytes.
(RTF)

**S2 Text. Bacterial endophyte IALR1619 effect on cucumber *Pythium ultimum* pathosystem.** (Table 1) Shoot dry weight of Cucumber plants treated with and without endophyte IALR1619. (Table 2) Dry shoot weight comparison for IALR1619 inoculated and non-inoculated cucumber plants. (Tables 3 and 4) Chi Square test to ascertain the effect of IALR1619 on cucumber seedlings inoculated with *Pythium*.
(RTF)

**S3 Text. Raw data and statistical analysis for the 1st hydroponic lettuce experiment.**
(RTF)

**S4 Text. Raw data and statistical analysis for the 2nd hydroponic lettuce experiment.**
(RTF)

**S5 Text. Raw data and statistical analysis for the hydroponic lettuce in perlites pots.**
(RTF)

## Author Contributions

**Conceptualization:** B. Sajeewa Amaradasa, Scott Lowman.

**Data curation:** B. Sajeewa Amaradasa.

**Formal analysis:** B. Sajeewa Amaradasa.

**Funding acquisition:** Chuansheng Mei, Scott Lowman.

**Investigation:** B. Sajeewa Amaradasa, Yimeng He, Robert L. Chretien, Mitchell Doss.

**Methodology:** B. Sajeewa Amaradasa.

**Project administration:** B. Sajeewa Amaradasa.

**Resources:** Chuansheng Mei, Scott Lowman.

**Supervision:** B. Sajeewa Amaradasa.

**Validation:** B. Sajeewa Amaradasa.

**Visualization:** B. Sajeewa Amaradasa.

**Writing – original draft:** B. Sajeewa Amaradasa.

**Writing – review & editing:** Chuansheng Mei, Yimeng He, Robert L. Chretien, Tim Durham, Scott Lowman.

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
