## [Decision Letter · Decision Letter 0]

5 Nov 2023

PONE-D-23-34185Biocontrol potential of endophytic *Pseudomonas* strainIALR1619 against two *Pythium* species in cucumber and hydroponic lettucePLOS ONE

Dear Dr. Amaradasa,

Thank you for submitting your manuscript to PLOS ONE. After careful consideration, we feel that it has merit but does not fully meet PLOS ONE’s publication criteria as it currently stands. Therefore, we invite you to submit a revised version of the manuscript that addresses the points raised during the review process.

We look forward to receiving your revised manuscript.

Kind regards,

Eugenio Llorens

Academic Editor

PLOS ONE

Journal Requirements:

   "This research was partially funded by USDA Specialty Crop Block Grant Program 2021B-570 for Utilizing Endophytes to Promote Hydroponic Vegetable Growth and Increase Profitability."

   "Initials of the authors who received the award :CM, SL

Grant number: 2021B-570

Funder: USDA Specialty Crop Block Grant Program. 

URL: https://www.ams.usda.gov/services/grants/scbgp

The funders Did not play any role in the study design data collection and analysis, decision to publish, or preparation of the manuscript."

Reviewers' comments:

Reviewer's Responses to Questions

**Comments to the Author**

1. Is the manuscript technically sound, and do the data support the conclusions?

Reviewer #1: Yes

Reviewer #2: Partly

Reviewer #3: Yes

2. Has the statistical analysis been performed appropriately and rigorously? 

Reviewer #1: No

Reviewer #2: Yes

Reviewer #3: Yes

3. Have the authors made all data underlying the findings in their manuscript fully available?

Reviewer #1: No

Reviewer #2: Yes

Reviewer #3: Yes

4. Is the manuscript presented in an intelligible fashion and written in standard English?

Reviewer #1: Yes

Reviewer #2: Yes

Reviewer #3: Yes

5. Review Comments to the Author

Reviewer #1: Dear Corresponding Author

I checked your paper and I have some comments to improve it for final publication:

1) Please draw a phylogenetic tree for both Pythium and bacterial isolates that you used in your study.

2) You need to have a bar-chart with statistics on growth rate of fungi against bacterial isolates.

3) You need to checke the normality of all of your data. If they are not normal you cannot analyze them with ANOVA.

4) How you identify the VOCs? Please add in-vitro photos for that. For examine the VOCs you will need to use two-partitioned-plates or fungal plate on bacterial palte. Please describe it.

5) Why you do not use LC-MS and just go ahead for GC-MS?

Regards

Reviewer #2: The study focuses on the effectiveness of the bacterial endophyte IALR1619 in protecting cucumber seedlings against the soil-borne pathogen P. ultimum. Results show a significant increase in survival rates and long-term effects of the endophyte. Furthermore, experiments on hydroponic lettuce demonstrate improved shoot fresh weight with IALR1619, suggesting its potential as a biocontrol agent against Pythium species in cucumber and hydroponic lettuce systems. I suggest the following changes:

- Keywords should not repeat words from the title.

- Unit style is incorrect (it should be given in a form of product, e.g. mg·L-1).

- Throughout the text, the authors are incorrectly using the term ‘variety’ instead of ‘cultivar’ and ‘sterilization’ instead of ‘disinfection’.

- Names of cultivars should be written in ‘’ (‘Cristabel’).

- Why are you writing about succulent plants in the Introduction?

- Please, explain the novelty of your study.

- Reference for the LB medium is missing. Who was the producer?

- Use mL not ml

- Some symbols are use incorrectly, i.e. x instead of ×.

- Lux is an outdated unit. Please provide the PPFD value.

- Fig. 3 lacks scale bars.

- In graphs, name both axes.

- Conclusions are missing.

- Reference #2: what year?

- Minor punctuation corrections are needed.

Reviewer #3: What about the keyword?

It is better to added good recommendation in abstract

the references that you used are little bit old you should update

Also write your objective clear

Make sure that all scientific names in the References list are italics.

All tables and Fig must be self-explanatory. This very important because you did not mentioned details about the experiment on the Fig

6. PLOS authors have the option to publish the peer review history of their article (what does this mean?). If published, this will include your full peer review and any attached files.

Reviewer #1: No

Reviewer #2: No

Reviewer #3: **Yes: **Kamal Abo-Elyousr

---

## [Author Response · Author response to Decision Letter 0]

19 Dec 2023

Reviewer #1: Dear Corresponding Author

I checked your paper and I have some comments to improve it for final publication:

1) Please draw a phylogenetic tree for both Pythium and bacterial isolates that you used in your study. 

We thank you for your suggestion. A phylogenetic tree for the isolates in this study has been added to the manuscript. A tree for two Pythium isolates was not prepared since they were not a part of the study. These isolates were verified as Pythium using macro and micromorphology and their symptoms on experimented plants resembled Pythium blight. There are similar peer reviewed biocontrol publications without phylogenetic trees for pathogens used.

2) You need to have a bar-chart with statistics on growth rate of fungi against bacterial isolates. 

As described in the manuscript, we screened >350 endophytes rapidly against a Pythium isolate and selected the 2 best endophytes for testing further in vitro and in greenhouse. First, a dual culture assay was done to ascertain biocontrol ability of these strains in vitro using two Pythium species followed by greenhouse studies. It was not our intention to compare these two using statistics. Therefore, we feel it is not necessary to include a bar chart to show the inhibition of Pythium isolates by bacterial strains in the main body of manuscript. However, a Supplementary Bar-chart figure was added (S1 Fig). In the manuscript, to show statistics, we added standard deviation value after % growth inhibition caused by bacterial endophytes. 

3) You need to checke the normality of all of your data. If they are not normal you cannot analyze them with ANOVA. 

The normality of data sets was checked using Shapiro-Wilk test using Proc Univariate procedure in SAS. The Dry shoot and root weights of Experiment 3 were not normal. Therefore, a non-parametric test called Kruskal- Wallis was used to analyze these two data sets. Other data sets were either normal or became normal when extreme observations were removed. The manuscript was edited accordingly.

4) How you identify the VOCs? Please add in-vitro photos for that. For examine the VOCs you will need to use two-partitioned-plates or fungal plate on bacterial palte. Please describe it. 

We used a fungal plate on a bacterial plate to ascertain if bacterial VOCs have any inhibition on Pythium isolates. This is clearly stated under the heading “Antifungal activity of exudates and VOCs of strain IALR1619” in methods. The assay in our manuscript is named “a two-sealed-bottom-plate assay”. The original manuscript had a photo of the effect of bacterial VOCs on Pythium (see Fig 7 in revised manuscript). The photo shows only the mycelial inhibition and there is no bacterial plate.

5) Why you do not use LC-MS and just go ahead for GC-MS? 

We used GC-MS method because VOCs are emitted as gas and not liquid.

Regards

Reviewer #2: The study focuses on the effectiveness of the bacterial endophyte IALR1619 in protecting cucumber seedlings against the soil-borne pathogen P. ultimum. Results show a significant increase in survival rates and long-term effects of the endophyte. Furthermore, experiments on hydroponic lettuce demonstrate improved shoot fresh weight with IALR1619, suggesting its potential as a biocontrol agent against Pythium species in cucumber and hydroponic lettuce systems. I suggest the following changes:

- Keywords should not repeat words from the title.

This journal does not include keywords in its articles. It is for online search only. That is why we also included some words from the title in keywords.

- Unit style is incorrect (it should be given in a form of product, e.g. mg·L-1) .

It was corrected.

- Throughout the text, the authors are incorrectly using the term ‘variety’ instead of ‘cultivar’ and ‘sterilization’ instead of ‘disinfection’. 

After some research it was found the lettuce names were cultivars and not varieties. “Variety” was changed to “cultivar” for lettuce. But cucumber Marketmore 76 is a variety. We did not change that. ‘Sterilize’ can be used as a synonym to ‘disinfect’. This word is used only twice. We added ‘surface sterilized’ to avoid any confusion in the first instance. In the second instance, sterilize’ is used to describe disinfestation of grains by autoclaving. We believe it is correct. 

- Names of cultivars should be written in ‘’ (‘Cristabel’). 

Inverted commas were added to all cultivar names.

- Why are you writing about succulent plants in the Introduction? 

Removed words “succulent plants” from the sentence.

- Please, explain the novelty of your study. 

The purpose/novelty of the study is mentioned in the introduction. The use of fungicides to manage disease has led to multiple environmental externalities, including resistance development, pollution, and non-target mortality. Growers have limited options as legacy chemistries are withdrawn from the market. Moreover, fungicides are generally labeled for traditional soil-based production, and not for liquid culture systems. Pythium is a common and devastating disease in many crops. Our study was an attempt to find biocontrol agents to control Pythium in hydroponic lettuce greenhouse cucumber. Biocontrol methods are more sustainable and there is a huge market in future. This can be found in the manuscript.

- Reference for the LB medium is missing. Who was the producer?

Manufacturer details added.

- Use mL not ml

ml changed to mL

- Some symbols are use incorrectly, i.e. x instead of ×.

Corrected

- Lux is an outdated unit. Please provide the PPFD value. √

Changed the units.

- Fig. 3 lacks scale bars. 

Thank you for highlighting the lack of scale bar in Fig3 (Fig 4 in revised manuscript). This photo was included to show relative sizes of representative plants from different treatments. We don’t have a photo with a ruler, and we don’t think not having a scale bar compromises understanding the growth differences. 

- In graphs, name both axes. 

We named both axes of Fig 5 graph. However, there are PLOS ONE papers similar to our original graph without a distinct name for X axis (Fig 4 original manuscript) (ex. https://journals.plos.org/plosone/article?id=10.1371/journal.pone.0286285 ). The figure captions have been changed to make figures more self-explanatory. 

- Conclusions are missing. 

A concluding statement was added to the end of discussion.

- Reference #2: what year? 

The year is 2020 and it is already included

- Minor punctuation corrections are needed.

We have corrected the language to our best.

Reviewer #3: 

What about the keyword? 

The question is not clear. Keywords are added as a requirement of the submission process although they don’t appear in the article. 

It is better to added good recommendation in abstract

A summary statement was added to the abstract.

the references that you used are little bit old you should update

References were updated.

Also write your objective clear

The objectives of the study were added at the end of the introduction. 

Make sure that all scientific names in the References list are italics. 

Corrected.

All tables and Fig must be self-explanatory. This very important because you did not mentioned details about the experiment on the Fig

Where necessary, the Table and Figure captions were added with more content to make them self-explanatory.

---

## [Decision Letter · Decision Letter 1]

28 Dec 2023

PONE-D-23-34185R1Biocontrol potential of endophytic *Pseudomonas* strain IALR1619 against two *Pythium* species in cucumber and hydroponic lettucePLOS ONE

Dear Dr. Amaradasa,

Thank you for submitting your manuscript to PLOS ONE. After careful consideration, we feel that it has merit but does not fully meet PLOS ONE’s publication criteria as it currently stands. Therefore, we invite you to submit a revised version of the manuscript that addresses the points raised during the review process.

Please, consider to clarify the comments suggested by the reviewers in this second round. Moreover, please, clarify the procedure to sterilize  the seeds by autoclave, and if the germination of seeds is affected. 

We look forward to receiving your revised manuscript.

Kind regards,

Eugenio Llorens

Academic Editor

PLOS ONE

Reviewers' comments:

Reviewer's Responses to Questions

**Comments to the Author**

1. If the authors have adequately addressed your comments raised in a previous round of review and you feel that this manuscript is now acceptable for publication, you may indicate that here to bypass the “Comments to the Author” section, enter your conflict of interest statement in the “Confidential to Editor” section, and submit your "Accept" recommendation.

Reviewer #1: (No Response)

Reviewer #2: All comments have been addressed

Reviewer #3: All comments have been addressed

2. Is the manuscript technically sound, and do the data support the conclusions?

Reviewer #1: Partly

Reviewer #2: Yes

Reviewer #3: Yes

3. Has the statistical analysis been performed appropriately and rigorously? 

Reviewer #1: No

Reviewer #2: Yes

Reviewer #3: Yes

4. Have the authors made all data underlying the findings in their manuscript fully available?

Reviewer #1: No

Reviewer #2: Yes

Reviewer #3: Yes

5. Is the manuscript presented in an intelligible fashion and written in standard English?

Reviewer #1: Yes

Reviewer #2: Yes

Reviewer #3: Yes

6. Review Comments to the Author

Reviewer #1: Dear Corresponding Author

When a reviewer has some comments, you have to do all of them into the paper. You have to do below comments and please consider that you cannot identify which comments are necessary. Because all of them are important. I do not want to reject your paper and then I revised your paper as "major revisions" again:

1) Please draw a phylogenetic tree for Pythium isolate(s) that you used in your study.

2) You need to have a bar-chart with statistics on growth rate of fungi against bacterial isolates.

3) About phylogenetic tree of bacterial isolates, you have 3 isolates that 2 of them are belonging to Pseudomonas but one of them is just completely separated from Pseudomonas and grouped with Enterobacter. In the title you just add IALR1619 for your bacterial strain. Could you please describe us why you add othe 2 strains?

4) About IALR1619 strain, it was completely separated from Pseudomonas gessardii with highly support (97% bootstrap). Is it a new species you think? I think you need to add more reference sequences from GenBank to more clarify the tree.

5) About GC-MS graphs, you need to add a table about the compounds that they are identified in you study.

Regards

Reviewer

Reviewer #2: Despite the claims of the authors, disinfection and sterilization are not the same. Sterilization is killing of all life forms. Alternatively, sterilization can mean castration of male flowers. According to the authors, "sterilize’ is used to describe disinfestation of grains by autoclaving". How is that even possible? If you autoclaved the grains, were they still alive?

Reviewer #3: (No Response)

7. PLOS authors have the option to publish the peer review history of their article (what does this mean?). If published, this will include your full peer review and any attached files.

Reviewer #1: No

Reviewer #2: No

Reviewer #3: No

---

## [Author Response · Author response to Decision Letter 1]

23 Jan 2024

Dear editor and reviewers

We are submitting the 2nd round of revised manuscript titled “Biocontrol potential of endophytic Pseudomonas strain IALR1619 against two Pythium species in cucumber and hydroponic lettuce” after incorporating suggestions by Academic Editor Dr. Liezl Callo and reviewer1. 

Answer to your question “clarify the procedure to sterilize the seeds by autoclave, and if the germination of seeds is affected”.

 We sterilized the grains by autoclaving in order to prepare the grain inoculum and not for germination. 

We have included all data sets of this research and statistical analysis reports as Supporting Information files (S2 to S6).

A rebuttal letter addressing all questions and suggestions raised by reviewer1 is attached separately, except for a phylogenetic tree for Pythium isolates used in this study. The academic editor Dr. Liezl Callo agreed it is not necessary/mandatory to have a phylogenetic tree of the pathogen in this type of studies. 

We have checked the quality of our Figures using PACE and they all met PLOS requirements.

Sincerely,

Sajeewa Amaradasa

---

## [Decision Letter · Decision Letter 2]

26 Jan 2024

Biocontrol potential of endophytic *Pseudomonas* strain IALR1619 against two *Pythium* species in cucumber and hydroponic lettuce

PONE-D-23-34185R2

Dear Dr. Amaradasa,

We’re pleased to inform you that your manuscript has been judged scientifically suitable for publication and will be formally accepted for publication once it meets all outstanding technical requirements.

Kind regards,

Eugenio Llorens

Academic Editor

PLOS ONE

Additional Editor Comments (optional):

Reviewers' comments:

Reviewer's Responses to Questions

**Comments to the Author**

1. If the authors have adequately addressed your comments raised in a previous round of review and you feel that this manuscript is now acceptable for publication, you may indicate that here to bypass the “Comments to the Author” section, enter your conflict of interest statement in the “Confidential to Editor” section, and submit your "Accept" recommendation.

Reviewer #1: All comments have been addressed

Reviewer #2: All comments have been addressed

2. Is the manuscript technically sound, and do the data support the conclusions?

Reviewer #1: Yes

Reviewer #2: Yes

3. Has the statistical analysis been performed appropriately and rigorously? 

Reviewer #1: Yes

Reviewer #2: Yes

4. Have the authors made all data underlying the findings in their manuscript fully available?

Reviewer #1: Yes

Reviewer #2: Yes

5. Is the manuscript presented in an intelligible fashion and written in standard English?

Reviewer #1: Yes

Reviewer #2: Yes

6. Review Comments to the Author

Reviewer #1: (No Response)

Reviewer #2: I have read the response letter and checked the ms. The manuscript is now suitable for publication. All questions were answered.

7. PLOS authors have the option to publish the peer review history of their article (what does this mean?). If published, this will include your full peer review and any attached files.

Reviewer #1: **Yes: **Ali Chenari Bouket

Reviewer #2: No

---

## [Editor Report · Acceptance letter]

17 Feb 2024

PONE-D-23-34185R2 

PLOS ONE

Dear Dr. Amaradasa, 

I'm pleased to inform you that your manuscript has been deemed suitable for publication in PLOS ONE. Congratulations! Your manuscript is now being handed over to our production team.

Kind regards, 

on behalf of

Dr. Eugenio Llorens 

Academic Editor

PLOS ONE